# Leptomeningeal Carcinomatosis in a Patient with Pancreatic Cancer: A Rare Phenomenon?

**DOI:** 10.3390/medicines9070039

**Published:** 2022-07-12

**Authors:** Taichi Sayanagi, Yumiko Ohishi, Makoto Katayama, Ryota Tamura

**Affiliations:** 1Department of Neurosurgery, Kawasaki Municipal Hospital, Kawasaki City 210-0013, Japan; taichisayanagi38@gmail.com (T.S.); makoto.katayama@gmail.com (M.K.); 2Department of Neurosurgery, Keio University School of Medicine, Tokyo 160-8582, Japan; ysyosk2010028@yahoo.co.jp

**Keywords:** chemo-resistant tumor, large-molecule systemic chemotherapies, leptomeningeal carcinomatosis, pancreatic cancer

## Abstract

Complication of leptomeningeal carcinomatosis (LMC) is critical. It causes rapid neurological deterioration, and subsequently, discontinuation of the ineffective treatment even in body tumor dormancy. Large molecular chemotherapeutic agents that are unlikely to penetrate the CSF space, are more likely to not treat LMC, typically in chemo-sensitive tumors. With the introduction of novel regimens, significant advances in overall survival have been observed even in formerly chemo-resistant tumors, such as pancreatic cancer. Although such cases are still rare, the number of pancreatic cancer patients complicated with LMC are increasing, and this therefore needs more recognition. A 49-year-old woman was diagnosed with stage IVa pancreatic cancer. She underwent surgery, and subsequent adjuvant chemotherapy. After three lines of chemotherapy over a 3-year period, where the body disease remained dormant, the patient was complicated by LMC. The diagnosis was made 4 months after the onset of headache. The patient received intrathecal methotrexate treatment but succumbed shortly after treatment induction. Pancreatic cancer is still relatively chemo-resistant and is one of the least likely types of tumor to be complicated by LMC due to patients dying of the primary tumor. Advancements in treatments have led to a prolonged period of primary tumor control, but not in the CNS due to the poor penetration of chemo-agents to this site. The present case seems to be a typical result of modern era anti-cancer therapy. Therefore, we emphasize the necessity of earlier recognition of this complication so that we can initiate specific treatment targeting the CSF space, especially in this formerly chemo-resistant tumor in order to improve its prognosis.

## 1. Introduction

Pancreatic cancer has one of the highest mortality rates among all cancers. Despite advances in therapy, prognosis of pancreatic cancer remains poor. Most patients die because of local recurrence of the disease and/or liver failure due to tumor infiltration [1]. According to the American Cancer Society, the one-year relative survival rate is 20%, and the five-year rate is 7%. However, survival rates have been incrementally improving with the development of novel treatment regimens over the past ten to fifteen years.

Leptomeningeal carcinomatosis (LMC) is an uncommon and late complication, occurring in 5–8% of cases of solid tumors, including pancreatic cancer. This complication comes with a poor prognosis and limited treatment options [2]. The incidence of LMC has been increasing due to the improved survival rates of cancer patients. Improved systemic control of the disease and therapies that do not cross the blood-brain barrier have contributed to this increase [3]. The longer patients live with systemic cancer, the higher the chances of metastasis to the leptomeninges.

Prognosis remains grim in patients with LMC. If left untreated, the time from the diagnosis to death is 4–6 weeks. It is known that the Karnofsky Performance Status (KPS) at diagnosis serves as an LMC prognostic factor and that the rate of death increases 19% per 10-unit decrease in KPS [4]. Additionally, LMC-related symptoms tend to be initially underdiagnosed in cancer patients due to the low specificity of the clinical manifestations, and also because healthcare professionals are not familiar with the characteristics and therefore do not consider the option of LMC being present. This leads to a delay in diagnosis and consequently, to a poor prognosis. Therefore, the early diagnosis of LMC is crucial in all cancer types with which it is associated. For early diagnosis, LMC must be widely comprehended by healthcare providers. To this end, studies about this condition are warranted, as knowledge of the subject remains limited. In line with this, we report a case of LMC in a patient with pancreatic cancer, corresponding to an increase in LMC prevalence.

## 2. Case Report

A 49-year-old woman with a history of jaundice that had persisted for 10 days presented to our outpatient clinic. Computed tomography (CT) of the chest and abdomen revealed a pancreatic head mass, as well as dilation of the pancreatic, common, and intrahepatic bile ducts. No metastases or lymphadenopathy was observed (Figure 1A).

Laboratory data demonstrated an elevation in total bilirubin (4.3 mg/dL; normal range, 0.2–1.0 mg/dL), aspartate aminotransferase (394 U/L; normal range, 0–35 U/L), and alanine aminotransferase (630 U/L; normal range, 0–35 U/L) levels. On examining the serum levels of tumor markers, normal carcinoembryonic antigen (CEA) (3.0 ng/mL; normal range, −5.0 ng/mL) and elevated carbohydrate antigen 19-9 (CA19-9) (269 kU/mL; normal level, −37 kU/mL) levels were observed as shown in Table 1. The patient underwent pancreaticoduodenectomy, and a pathological diagnosis of adenocarcinoma stage IVa (pT3N2M0) was made. Tegafur, gimeracil, oteracil potassium (TGOP) (120 mg/day, 4 weeks on/2 weeks off) was administered as adjuvant chemotherapy.

One year later CA19-9 elevation was noted. She was diagnosed with tumor recurrence, and the chemotherapy regimen was changed to combined TGOP (120 mg/day, on alternate days for 4 weeks) and gemcitabine (GEM), which was started at a dose of 875 mg/m^2^/week, 3 weeks on/1 week off. CT scans obtained 3 months later did not show any obvious findings of recurrence, but CA19-9 elevation was observed again. Indicative of a second recurrence, we changed the regimen to gemcitabine (625 mg/m^2^/week) and nab-paclitaxel (albumin-bound paclitaxel particles; 100 mg/m^2^/week).

Two months later, a follow-up truncal CT revealed metastases in the hilar and mediastinal lymph nodes. Therefore, a modified FOLFIRINOX (mFOLFIRINOX) regimen (oxaliplatin 85 mg/m^2^, levofolinate calcium 200 mg/m^2^, and irinotecan 150 mg/m^2^ every two weeks) was started as fourth-line chemotherapy. Around this time, the patient started to complain of headaches, which were initially thought to be a minor side effect of the chemotherapy.

Three months later, an abdominal CT scan was performed and a reduction in the size of the lymph node metastasis was noted. There were no findings indicative of postoperative recurrence.

Four months after the initiation of the mFOLFIRINOX regimen, she was admitted to the hospital for severe headache and nausea. Under a suspected diagnosis of LMC, her cerebrospinal fluid (CSF) was collected and the opening pressure was found to be elevated (25 cmH_2_O). CSF examination revealed 11 cells/3 fields, with normal protein and high glucose levels (20 mg/dL and 92 mg/dL, respectively). CSF cytology was negative, but an increase in CSF CEA level was noted (10.7 ng/mL). Symptomatic treatment was administered for the next two weeks, after which the patient was referred to the neuro-oncology department. Contrast-enhanced, T1-weighted magnetic resonance imaging (MRI) revealed leptomeningeal enhancement along the right parietal sulci (Figure 1B). CSF was obtained for the second time, and a small number of atypical cells were noted on cytological examination. Based on the clinical symptoms and CSF and MRI findings, the patient was diagnosed with LMC. Subsequently, an Ommaya reservoir was implanted in the anterior horn of the lateral ventricle to initiate intrathecal (IT) chemotherapy. CSF obtained intraoperatively from the ventricle contained malignant cells. On the day after Ommaya reservoir implantation, IT chemotherapy with methotrexate (10 mg) and dexamethasone (6.6 mg) was started. However, the consciousness of the patient deteriorated shortly after the initiation of IT chemotherapy, and she died two days after the implantation surgery. Autopsy revealed LMC metastases, without any evidence of recurrence at the primary lesion site or regional lymph nodes (Figure 1C,D).

## 3. Discussion

We here report a case of pancreatic cancer associated with LMC, a distant aggressive complication that is life threatening and rare. Pancreatic and biliary cancers are locally aggressive and chemo-resistant; hence, most patients succumb to the progression of their systemic disease prior to developing central nervous system (CNS) metastases [5]. Nonetheless, pancreatic cancers respond to treatment regimens, such as combinations of fluorouracil, irinotecan, oxaliplatin, and leucovorin, as well as gemcitabine plus nab-paclitaxel [6,7], resulting in improvement in the patient’s overall survival [8,9]. We chose the four treatment regimens for LMC: TGOP, TGOP + GEM, GEM + nab-paclitaxel and mFOLFIRINOX, according to the Clinical Guidelines for Pancreatic Cancer [10,11].

Currently, there are only fourteen reports of pancreatic cancer complicated by LMC in the literature [3,5,6,12,13,14,15,16,17,18,19,20,21,22]. Among those, five cases involved LMC after cancer onset, a rare presentation that is likely to remain so [5,6,17,19,20]. In contrast, LMC was the only active malignancy after the initiation of chemotherapy in the other five cases. The latter describes the case presented in the current report. To our knowledge, our case is different from the other five cases in that the patient is of Asian ethnicity. The course of the disease is similar to what has been observed in patients with HER2-positive breast cancer treated with the large molecule Trastuzumab, in which malignant breast cancer cells in the CSF were refractory to systemic therapy because this agent does not cross the blood-brain barrier [22].

The incidence of LMC in solid tumors has been increasing because of improved systemic chemotherapy and consequently longer survival rates in breast and lung cancers [23,24,25,26]. The large-molecule systemic chemotherapies that are typically used to control systemic disease cannot permeate the blood-brain barrier, which creates a sanctuary site where malignant cells are shielded from the effect of the therapy. Thus, malignant cells persist in the brain parenchyma and CSF, and thus cause brain metastases and LMC [27]. Clinically, the longer the patient responds to chemotherapeutic agents, the more likely it is that sites that cannot be reached by chemotherapeutic agents will be affected by the primary tumor.

Based on our findings, we would like to make clinicians aware that LMC may become more prevalent because of prolonged survival and to emphasize the need to recognize this devastating complication better in order to expedite its diagnosis to more accurately treat the patient with intrathecal therapy and radiation. We particularly aim to educate the pancreatic oncology community. Our case indicates that patients who respond well to systemic therapy should be carefully monitored for neurologic symptoms such as headaches and altered mental status. The early diagnosis of LMC is crucial, as neurological deficits caused by LMC are largely irreversible, greatly compromising patient survival.

In our case, although we underwent surgery and started intrathecal treatment immediately, the patient died within two days of surgery. The patient was referred to our department at a terminal stage after possibly more than several weeks from LMC onset. Thus, we believe the death was natural course and inevitable.

To the best of our knowledge, this is the first report to call attention to the increasing risk of LMC complication with prolonged overall survival rates resulting from current chemotherapies. We have reported a case of LMC in a patient with pancreatic cancer and discussed the rationale for awareness of such cases. A large-scale observational study of patients with pancreatic cancer is needed to confirm whether the number of LMC cases that derive from pancreatic cancer is actually increasing, and to identify LMC earlier when patients do not yet have devastating neurological symptoms, so that adequate treatment is administered sooner to prolong life with optimal quality.

## Figures and Tables

**Figure 1 medicines-09-00039-f001:**
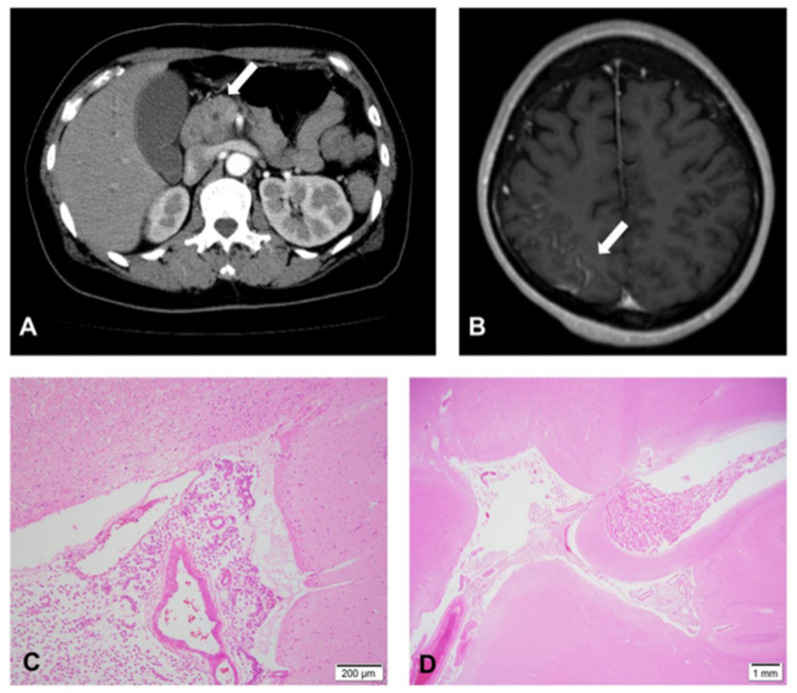
Characteristics of the pancreatic cancer case and associated leptomeningeal carcinomatosis. (**A**) Computed tomography of the abdomen showing the primary site of the pancreatic tumor, as indicated by the arrow. (**B**) Axial contrast-enhanced T1-weighted magnetic resonance imaging showing tumor cells infiltrating the sulci of the right parietal lobe. (**C**,**D**) Hematoxylin-eosin-stained histopathologic specimens obtained during autopsy showing a thin layer of infiltrated meninges surrounding the brain.

**Table 1 medicines-09-00039-t001:** Laboratory data of patients at different stages of chemotherapies.

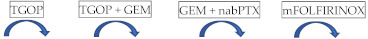
Time from Diagnosis	0 Months	15 Months	19 Months	21 Months	24 Months
Events	Diagnosis	1st recurrence	2nd recurrence	3rd recurrence	Before Death
T-Bil (mg/dL)	4.3	1.0	0.60	0.40	0.20
AST (U/L)	394	36	30	19	54
ALT (U/L)	630	37	33	22	62
CEA (ng/mL)	3.0	9.6	12.4	135.2	20.4
CA19-9 (kU/mL)	269	460	13,240	50,000	16,000

ALT: alanine aminotransferase, AST: aspartate aminotransferase, CEA: carcinoembryonic antigen, CA19-9: carbohydrate antigen 19-9, T-Bil: total bilirubin, TGOP: tegafur, gimeracil, oteracil potassium, GEM: gemcitabine, nabPTX: nab-paclitaxel, mFOLFIRINOX: modified FOLFIRINOX.

## Data Availability

Not applicable.

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
