# Peer review of "Leptomeningeal Carcinomatosis in a Patient with Pancreatic Cancer: A Rare Phenomenon?"

_medicines, 2022, doi:10.3390/medicines9070039_

Round 1
Reviewer 1 Report
Dear authors,
this is a very interesting subject which in all cancer types is a metastatic disease pattern of major concern because reaching the LM disease is as you stated extremely difficult. In most patients therefore, esp. in the relapse setting this is one of the reasons the patients die of their disease.
Therefore early diagnosis and more importantly effective treatment strategies or drug delivery strategies are needed to improve patient outcome.
The paper needs a thorough rewrite to improve its impact. I would therefore suggest and debate the following improvements, and on top of that a rewrite on the English language by a native speaker.
Remarks regarding abstract:
Line 12 – why treatment withdrawal? LMC compromises survival, it is not treatment withdrawal, it is stopping ineffective treatment
Line 12-13 – I understand what the authors tried to explain, however do not agree. Due to the blood brain barrier (BBB) and tight junctions most agents cannot pass the BBB, this does not “give rise” to LMC, this does not treat LMC. There is a difference here. This should be rewritten. Furthermore the term esp in chemosensitive tumours should be explained. I think the authors mean to say that the tumour might have responded if the agent was able to enter the tumour, describe this as such.
Line 17 – should “by” not be “with”
Line 19 – change “whereas” to “where”
Line 21 – intrathecal chemotherapy with what agent? Please describe
Line 21 – I assume you mean succumbed/died of disease, not expired?
Line 22 – it is debated later, but I am of the opinion that you do not see LMC much in this tumour group because patients die of the primary tumour before they can metastasize to the brain. I would also mention that here
Line 23 – “accumulation” is not proper English in this context
Line 24 – “ill penetration” is not proper English in this context
Line 24 – I am not sure about the term “body tumour dormancy”, would primary tumour control not be a better term?
Line 26 – delete “the” before earlier
Line 25-27 – the argument made at the end feels counter intuitive. There is earlier recognition (to save life? Prolong disease control? In a patient with a tumor responding to chemo or novel agents (because there is longer survival), but how to save the patient then with earlier recognition, because as stated it does not respond to the given treatment (because the LMC originates despite treatment). Please rewrite
Line 32 – delete “major”
Line 39 – delete “However”, start sentence with “This”
Line 48 – delete “with metastatic disease”
Line 50 – not only “low clinical manifestation” most likely also because the health care professional is not familiar with the characteristics and therefore does not consider the option of LMC being present
Line 50 – “to a poor prognosis” with earlier diagnosis would there be a better prognosis with current treatment strategies?
Line 68 and 70 – what are TS-1 and S-1?
Line 73 – delete “indicative”
Line 74 – change “under the diagnosis of “ into “Indicative of”
Line 84 – if as stated the prognosis for LMC is 4-6 weeks (line 46, and this is the reality), did the LMC in the patient not respond to administered treatment? Because if I understand the timeline well there were headaches well before (line 79) and LMC was discovered later (4 months?)
Line 89 – can the authors speculate, or much better have evidence for the sensitivity of increased CSF CEA level?
Line 98 – out of curiosity, why was the choice MTX, what is the MTX sensitivity of pancreatic tumours?
Line 111 – should “locally” not be “distant”? because it is metastatic disease
Line 118 – “thee” typing error
Line 124 – what “malignant cells”? should in the CSF be added?
Line 125 – not only solid but also braintumour patients where there is local disease control, but insufficient CSF penetration to access the LMC, because you can enter the primary brain tumour (disrupted BBB) and thus have local control, but not the CSF (LMC is predominantly treated by diffusion, that is why you administer intra-CSF), thus not (line 127) “particularly” in breast and lung cancer, but potentially/most likely in a lot of tumours with increased local control
Line 129 – delete “likely”
Line 131 – replace “in” with “on”, and “resulting in the development of” with “and thus”
Line 132/134 – not proper English rewrite sentence to make the intention of the sentence clear, affected of what? (I understand what you mean, but I have to come up with the suggestion myself)
Line 135/137 – this is copy of statement line 126/128
Line 137/139 – also double, and if you would like a statement and thus repeat to increase impact, add: more prevalent because of prolonged survival, and expedite diagnosis to more accurately treat (how?). Or if this was implied by repeating line 135/137 it misses in line 135 that the patients until recently succumbed to their local disease (because of inadequate treatment)
Line 143 – does it compromise “further treatment” or patient survival?
Line 149 – is the aim to know “whether …. Is increasing” or to identify earlier when patient does not have that many (devastating) neurological symptoms yet and proper/adequate treatment directed at the LMC is administered asap to prolong life with optimal quality or potentially cure (although I do not think we have reached that state yet, or will reach that state in the near future).
Best wishes,
the reviewer
Author Response
We are very grateful to the reviewers for their insightful comments and suggestions, which would undoubtedly help us to improve our manuscript immensely. As indicated in the responses below, we have taken all their comments and suggestions into account when generating the revised version of the manuscript. Responses to the reviewers’ comments appear after the arrows, in blue text.
Reviewer #1:
Line 12 – why treatment withdrawal? LMC compromises survival, it is not treatment withdrawal, it is stopping ineffective treatment
→
I agree with your opinion. I have changed the term to “discontinuation of ineffective treatment” in the manuscript.
Line 12-13 – I understand what the authors tried to explain, however do not agree. Due to the blood brain barrier (BBB) and tight junctions most agents cannot pass the BBB, this does not “give rise” to LMC, this does not treat LMC. There is a difference here. This should be rewritten.
→
Thank you for your comments.
I have rewritten “to give rise” to “to not treat”.
Furthermore the term esp in chemosensitive tumours should be explained. I think the authors mean to say that the tumour might have responded if the agent was able to enter the tumour, describe this as such.
→
I have written “typically in chemo-sensitive tumors” to emphasize that LMC are more likely to occur in patients with chemo-sensitive tumors because they tend to survive longer, and therefore have more chance of intracranial lesions (such as LMC) where large molecular chemotherapeutic agents are not able to reach.
Line 17 – should “by” not be “with”
→
Thank you for your comments.
I have changed “by” to “with”.
Line 19 – change “whereas” to “where”
→
Thank you for your comments.
I have changed “whereas” to “where”.
Line 21 – intrathecal chemotherapy with what agent? Please describe
→
Thank you for your comments.
I have changed “intrathecal chemotherapy” to “intrathecal methotrexate treatment”.
Line 21 – I assume you mean succumbed/died of disease, not expired?
→
Thank you very much for pointing out. I have rewritten the phrase using “succumbed”
Line 22 – it is debated later, but I am of the opinion that you do not see LMC much in this tumour group because patients die of the primary tumour before they can metastasize to the brain. I would also mention that here
→
Thank you for your comments.
I have added the phrase.”due to patients dying of the primary tumor”.
Line 23 – “accumulation” is not proper English in this context
→
Thank you for finding my mistake.
I have rewritten the phrase.
Line 24 – “ill penetration” is not proper English in this context
→
Thank you for pointing out.
I have changed “ill” to “poor”.
Line 24 – I am not sure about the term “body tumour dormancy”, would primary tumour control not be a better term?
→
Thank you for your comments.
I have changed the term to “primary tumor control”.
Line 26 – delete “the” before earlier
→
Thank you for finding my mistake.
I have deleted the term.
Line 25-27 – the argument made at the end feels counter intuitive. There is earlier recognition (to save life? Prolong disease control? In a patient with a tumor responding to chemo or novel agents (because there is longer survival), but how to save the patient then with earlier recognition, because as stated it does not respond to the given treatment (because the LMC originates despite treatment). Please rewrite
→
Thank you for your comments.
I agree that the prognosis is poor even if LMC was diagnosed early in the course. However, there are specific treatments targeting the CSF space that are reported to prolong the overall survival rate and quality of life. In this context we believe that being aware of the complication is meaningful. I have added a phrase to be more specific.
Line 32 – delete “major”
Line 39 – delete “However”, start sentence with “This”
Line 48 – delete “with metastatic disease”
→
Thank you for your comments.
I have omitted these words from the manuscript.
Line 50 – not only “low clinical manifestation” most likely also because the health care professional is not familiar with the characteristics and therefore does not consider the option of LMC being present
→
I agree with your opinion. I have added the phrase “and also because the health care professional is not familiar with the characteristics and therefore does not consider the option of LMC being present ” in the manuscript.
Line 50 – “to a poor prognosis” with earlier diagnosis would there be a better prognosis with current treatment strategies?
→
Thank you for your comments.
I completely agree that the prognosis is poor even if the diagnosis is made early. However, there are reports saying that early diagnosis and therapy is critical[1]. Also, intrathecal chemotherapy and radiation could prolong survival[2].
Line 68 and 70 – what are TS-1 and S-1?
→
Thank you for your comments.
TS-1 and S-1 are both Tegafur, Gimeracil, Oteracil Potassium.
I have rewritten the phrase.
Line 73 – delete “indicative”
Line 74 – change “under the diagnosis of “ into “Indicative of”
→
Thank you for your comments.
I have changed the above.
Line 84 – if as stated the prognosis for LMC is 4-6 weeks (line 46, and this is the reality), did the LMC in the patient not respond to administered treatment? Because if I understand the timeline well there were headaches well before (line 79) and LMC was discovered later (4 months?)
→
Thank you for your comments.
I totally agree that the headache was present well before the diagnosis of LMC was made. As you have pointed out, we cannot rule out the possibility that the administered treatment was effective. However, the exact timing when LMC occurred cannot be known and there is a possibility that the disease progression was slower in this case.
Line 89 – can the authors speculate, or much better have evidence for the sensitivity of increased CSF CEA level?
→
Thank you for your comments.
Although the exact numbers are not shown, there is a paper showing that the sensitivity of tumor markers in the diagnosis of leptomeningeal carcinomatosis from solid cancers are high[3]. Although the study did not include any patient with LMC from pancreatic cancer, we speculate the sensitivity in our case would be close.
Line 98 – out of curiosity, why was the choice MTX, what is the MTX sensitivity of pancreatic tumours?
→
Thank you for your comments.
MTX, cytarabine, including liposomal cytarabine, or thioTEPA has been mainly used in intrathecal therapy. Due to the availability at our hospital, we chose MTX. As far as we are aware, the sensitivity of pancreatic tumors to MTX remains unclear.
Line 111 – should “locally” not be “distant”? because it is metastatic disease
Line 118 – “thee” typing error
→
Thank you for pointing out.
I have corrected the sections noted above.
Line 124 – what “malignant cells”? should in the CSF be added?
→
Thank you for pointing out.
I have added “breast cancer” and “in the CSF” in the manuscript.
Line 125 – not only solid but also brain tumour patients where there is local disease control, but insufficient CSF penetration to access the LMC, because you can enter the primary brain tumour (disrupted BBB) and thus have local control, but not the CSF (LMC is predominantly treated by diffusion, that is why you administer intra-CSF), thus not (line 127) “particularly” in breast and lung cancer, but potentially/most likely in a lot of tumours with increased local control
→
Thank you for your comments.
I have deleted “particularly” from the manuscript.
Line 129 – delete “likely”
Line 131 – replace “in” with “on”, and “resulting in the development of” with “and thus”
→
Thank you for pointing out.
I have corrected the above.
Line 132/134 – not proper English rewrite sentence to make the intention of the sentence clear, affected of what? (I understand what you mean, but I have to come up with the suggestion myself)
→
Thank you for your comments.
I have added “by the primary tumor” in the end of the sentence.
Line 135/137 – this is copy of statement line 126/128
→
Thank you for your comments.
I have omitted the sentence from the manuscript.
Line 137/139 – also double, and if you would like a statement and thus repeat to increase impact, add: more prevalent because of prolonged survival, and expedite diagnosis to more accurately treat (how?). Or if this was implied by repeating line 135/137 it misses in line 135 that the patients until recently succumbed to their local disease (because of inadequate treatment)
→
Thank you for your comments.
I have rewritten the sentence as below.
“Based on our findings, we would like to make clinicians aware that the phenomenon of LMC may become more prevalent because of prolonged survival and to emphasize the need to recognize this devastating complication better in order to expedite its diagnosis to more accurately treat the patient with intrathecal therapy and radiation.”
Line 143 – does it compromise “further treatment” or patient survival?
→
Thank you for your comment.
As you have pointed out, it is patient survival that is compromised. I have rewritten the phrase accordingly.
Line 149 – is the aim to know “whether …. Is increasing” or to identify earlier when patient does not have that many (devastating) neurological symptoms yet and proper/adequate treatment directed at the LMC is administered asap to prolong life with optimal quality or potentially cure (although I do not think we have reached that state yet, or will reach that state in the near future).
→
Thank you for your comment.
To convey the aim of the study, , I have rewritten the sentence as below.
“A large-scale observational study of patients with pancreatic cancer is needed to confirm whether the number of LMC cases that derive from pancreatic cancer is actually increasing, and to identify LMC earlier when patients don’t have devastating neurological symptoms yet so that adequate treatment is administered sooner to prolong life with optimal quality.”
References
- Grossman, S.A. and M.J. Krabak, Leptomeningeal carcinomatosis. Cancer Treat Rev, 1999. 25(2): p. 103-19.
- Sause, W.T., et al., Whole brain irradiation and intrathecal methotrexate in the treatment of solid tumor leptomeningeal metastases--a Southwest Oncology Group study. J Neurooncol, 1988. 6(2): p. 107-12.
- Corsini, E., et al., Intrathecal synthesis of tumor markers is a highly sensitive test in the diagnosis of leptomeningeal metastasis from solid cancers. Clin Chem Lab Med, 2009. 47(7): p. 874-9.

Reviewer 2 Report
Pancreatic cancer is hard to diagnose early, since pancreatic cancer signs and symptoms aren’t obvious. Because of this, the majority of these cancers are diagnosed after the disease has reached an advanced stage, when treatment options are limited.
Sayanagi et al., reported a 49-year old woman was diagnosed with stage IVa pancreatic cancer and developed complications of leptomeningeal carcinomatosis after receiving three lines of chemotherapy over a three year period. The present case seems to be a typical result of modern era anti-cancer therapy.
It is a nice written report and I have several comments below
- The authors should have a table of all information of the laboratory data to easily follow the number (Page 61-68). All this laboratory data was recorded after the patients received 3 lines of chemotherapy or at which stage of her disease, the authors should clarify in the case report.
- It seems that the CA19-9 level keeps elevated whenever the tests are performed. The author should explain why they emphasize CA19-9 in the case report and why they decided to combine S1 and gemcitabine for chemotherapy; then gemcitabine and nab-paclitaxel, finally FOLFIRINOX in the text.
- In the discussion part, could the author discuss the report case in this article that is different from the other 5 cases involving LMC after cancer onset?
- In page 140, the author mentioned that “Our case indicates that patients who respond well to systemic therapy should be carefully monitored for neurologic symptoms”. Could the authors explain more detail about this statement?
- What is the lesson that the authors learnt from this case report? Is there any suggestion for the potential therapeutic treatment for similar cases like this?
- Can the authors discuss potential reasons why the patient died after two days of surgery?
Author Response
We are very grateful to the reviewers for their insightful comments and suggestions, which would undoubtedly help us to improve our manuscript immensely. As indicated in the responses below, we have taken all their comments and suggestions into account when generating the revised version of the manuscript. Responses to the reviewers’ comments appear after the arrows, in blue text.
Reviewer #2:
Pancreatic cancer is hard to diagnose early, since pancreatic cancer signs and symptoms aren’t obvious. Because of this, the majority of these cancers are diagnosed after the disease has reached an advanced stage, when treatment options are limited.
Sayanagi et al., reported a 49-year old woman was diagnosed with stage IVa pancreatic cancer and developed complications of leptomeningeal carcinomatosis after receiving three lines of chemotherapy over a three year period. The present case seems to be a typical result of modern era anti-cancer therapy.
It is a nice written report and I have several comments below
- The authors should have a table of all information of the laboratory data to easily follow the number (Page 61-68). All this laboratory data was recorded after the patients received 3 lines of chemotherapy or at which stage of her disease, the authors should clarify in the case report.
→
Thank you for your comments. I have added a new figure which contains a table showing all the laboratory data at each event.
- It seems that the CA19-9 level keeps elevated whenever the tests are performed. The author should explain why they emphasize CA19-9 in the case report and why they decided to combine S1 and gemcitabine for chemotherapy; then gemcitabine and nab-paclitaxel, finally FOLFIRINOX in the text.
→
Thank you for your comments. We emphasized the CA19-9 in the case report, and chose the chemotherapy as above to comply with the Clinical Guidelines for Pancreatic Cancer [1].
- In the discussion part, could the author discuss the report case in this article that is differentfrom the other 5 cases involving LMC after cancer onset?
→
Thank you for your comments. All of the cases have unique courses, however only our case is different from the other 5 cases in that the patient is of Asian ethnicity. I have added a sentence to describe this point.
- In page 140, the author mentioned that “Our case indicates that patients who respond well to systemic therapy should be carefully monitored for neurologic symptoms”. Could the authors explain more detail about this statement?
→
Thank you for your comments. I have added “such as headaches and altered mental status” at the end of the sentence to be more specific.
- What is the lesson that the authors learnt from this case report? Is there any suggestion for the potential therapeutic treatment for similar cases like this?
→
Thank you for your comments.
We have learned from the case that we should be more aware of LMC even in cancers thought to have low risk for LMC. There is a report which suggest that the combination of gemcitabine plus nab-paclitaxel is potentially effective in affected cerebrospinal fluid of pancreatic carcinoma patients [2], thus we might use this regimen in future cases.
- Can the authors discuss potential reasons why the patient died after two days of surgery?
→
Thank you for your comments.
The patient was referred to our department at a terminal phase, with possibly more than several weeks after LMC onset. Thus we believe the death was natural course and inevitable.
I have added a section to discuss this topic just before the last paragraph.
References
- Yamaguchi, K., et al., EBM-based Clinical Guidelines for Pancreatic Cancer (2013) issued by the Japan Pancreas Society: a synopsis. Jpn J Clin Oncol, 2014. 44(10): p. 883-8.
- Ceccon, G., et al., Leptomeningeal Carcinomatosis in a Patient with Pancreatic Cancer Responding to Nab-Paclitaxel plus Gemcitabine. Case Rep Oncol, 2020. 13(1): p. 35-42.
